# Flux Coupling and the Objective Functions’ Length in EFMs

**DOI:** 10.3390/metabo10120489

**Published:** 2020-11-28

**Authors:** Francisco Guil, José F. Hidalgo, José M. García

**Affiliations:** Grupo de Arquitectura y Computación Paralela, Universidad de Murcia, 30080 Murcia, Spain; fguil@um.es (F.G.); jmgarcia@um.es (J.M.G.)

**Keywords:** metabolic networks, linear programming, EFM, flux modes, pathways, systems biology

## Abstract

Structural analysis of constraint-based metabolic network models attempts to find the network’s properties by searching for subsets of suitable modes or Elementary Flux Modes (EFMs). One useful approach is based on Linear Program (LP) techniques, which introduce an objective function to convert the stoichiometric and thermodynamic constraints into a linear program (LP), using additional constraints to generate different nontrivial modes. This work introduces FLFS-FC (Fixed Length Function Sampling with Flux Coupling), a new approach to increase the efficiency of generation of large sets of different EFMs for the network. FLFS-FC is based on the importance of the length of the objective functions used in the associated LP problem and the imposition of additional negative constraints. Our proposal overrides some of the known drawbacks associated with the EFM extraction, such as the appearance of unfeasible problems or multiple repeated solutions arising from different LP problems.

## 1. Introduction

Cellular metabolism is nowadays an active and fruitful field of research, partly because of its close connections to applications in biology and medicine, such as the analysis of different types of cancer and other diseases [1]. From a formal point of view, metabolism can be modeled as a network of reactions transforming metabolites into other metabolites. One of the key ways cellular metabolism can be understood is by analyzing these networks’ structural properties. Structural analyses are usually performed using graph-related techniques or constraint-based modeling. The latter term was introduced in [2,3]. In this approach, the network is modeled as a graph or hypergraph with several constraints to restrict the set of possible fluxes. These constraints can be stoichiometric (based on the quantities of metabolites involved in each reaction), thermodynamic (restricting the direction of some reactions), or regulatory (generally based on gene regulations of the network). The kind of restrictions imposed on a network depends on the problem of interest or the information about the metabolism that requires analysis.

Once the model has been fixed, only those fluxes that meet the restrictions are of interest. In stoichiometric and thermodynamic constraints, an admissible flux distribution is a set of reactions where all intermediate compounds are balanced and irreversible reactions run in the appropriate direction. This flux distribution is called a mode or a pathway [4].

The set of modes is typically infinite, so the objective is to find relevant finite subsets that could generate the full set of modes. Several strategies have been proposed to find such exciting subsets (see in [4,5,6]). Two of the most used are the so-called elementary modes that are minimal ones (see in [6]) and extreme pathways (see in [4]). As stated in [5], both definitions agree when all the network’s reactions are irreversible. It is well known that reversible reactions can be modeled using two irreversible ones (see in [7]), so in this paper we assume that all the reactions are irreversible. Therefore, we use the term minimal mode (referred to as Elementary Flux Mode (EFM)) in the sense of being minimal or extreme. EFMs have essential relationships with other key concepts in constraint-based modeling, such as blocked reactions (which are essentially the inconsistent enzyme subsets used in the description and implementation of the program Metatool [8]), flux coupling [9], or cut sets [10]. These connections can be used to determine, for example, if a particular set is a cut set for a given reaction after the set of all EFMs has been calculated [10].

Even though the number of different EFMs has an upper bound (see in [11,12]), this number is usually quite large. Therefore, sometimes the main interest is not to find the whole set of EFMs (due to its cardinality) but to construct a large set of different EFMs that can be used to study the key properties of the network (see in [13,14,15]).

Several strategies have been proposed to calculate all the EFMs of a network, such as techniques related to the double description method [16,17] or mixed-integer linear programming [18,19]. Both approaches are computationally demanding. A less expensive computation method is based on linear programming.

Linear program (LP) techniques are especially suitable in the construction of EFMs. LP methods are computationally efficient and many available libraries can be used. The approach’s fundamental idea is the use of linear optimization techniques to find large subsets of EFMs by imposing additional requirements. The starting point is the stoichiometric and thermodynamic constraints imposed on the reactions that define a polyhedral cone of solutions. To pose an optimization problem, a linear objective function based on the network’s different reactions has to be defined to be optimized. This objective function can use some or all of the network reactions. Depending on the number of reactions used, objective functions with different lengths are constructed, being the length the number of reactions that appear with non-zero coefficients in the definition of the objective function.

Besides, it is also possible to use additional constraints in LP optimization problems. These additional constraints are made by imposing those specific reactions that must be present (or absent) in the solution. Using different objective functions and imposing additional constraints, it is possible to define different linear programs whose solutions are the modes of the network. It is worth noting that choosing different functions or sets of constraints does not guarantee the obtention of different EFMs.

Previously, other authors have followed LP strategies to find EFMs in metabolic networks. Some well-known approaches are EFMEvolver [14], which is proposed to be used in conjunction with genetic algorithms, or treeEFM [20], proposed using with binary trees.

This paper proposes FLFS-FC (Fixed Length Function Sampling with Flux Coupling), a new approach to finding large sets of different EFMs for metabolic networks using LP methods. Our methodology is based on the study of the relationship between the length of the objective function and the rate of repeated solutions obtained. FLFS-FC focuses on the length of the objective functions used in conjunction with the use (in each step) of additional constraints that force the absence of some reactions randomly chosen from a suitable set. This set of reactions is obtained by choosing those that appear with higher frequencies in the first computed EFMs.

The main contributions of this paper are the following.
The relationship between the number of repeated solutions and the length of the objective functions used.The use of negative constraints as a means of improving the diversity of the solutions obtained.The development of a new proposal (FLFS-FC) to obtain large sets of EFMs. This proposal has efficiency rates far better than previously proposed ones.

As far as we know, this is the first appearance in the literature of this kind of study, and it serves to improve significantly the results obtained in previous attempts to obtain large sets of EFMs.

## 2. Results

In this study, we have used as evaluation platform a computer equipped with a double socket Cascade Lake Xeon Gold 6238 (44 cores) @ 2.2 GHz with 384 GB of RAM. The system runs on a CentOS Linux 7.5, running CPLEX 12.10 version from IBM and Python 3.6.8 version from Intel.

The selected metabolic reconstruction for our results is *iAF1260*, the reconstruction of the *E. coli* K-12 MG1655 organism [21]. Its stoichiometric matrix, once decoupled (that is, after replacing any reversible reaction by two irreversible ones), has 3234 reactions.

This network was previously used in [20] to compare the different efficiency rates obtained by EFMEvolver [14] and treeEFM [20] (two LP methods) by computing the first 2000 EFMs. In that paper, the target reactions (those forced to appear in the EFMs computed) were L-Lysine, L-Arginine, and L-Theronine.

We have reproduced those experiments to properly compare FLFS-FC with them. In our tests, the efficiency rates are computed as the mean of the rates obtained from 40 experiments, each used to compute 2000 EFMs containing the respective target reactions. Table 1 shows the efficiency rates obtained by EFMEvolver, treeEFM, and FLFS-FC. For the sake of completion, we have also added two columns showing the cardinality of the reaction sets $I^{\prime }$ and that of reaction sets *K* used in the negative additional constraint (denoted by |I${}^{\prime }$| and |K|). Notice that, as expected, the cardinality of *K* varies with the selected target. Finally, the last column shows the improvement ratio obtained using FLSL-FC compared to treeEFM according to the number of repeated EFMs.

It is worth noting that the efficiency rates obtained are relatively stable, with standard deviations for the efficiency rates being 0.0079, 0.0025, and 0.003137 for FLFS-FC and the reaction targets, L-Lysine, L-Thereonine, and L-Arginine, respectively.

As expected, efficiency rates worsen as larger sets of EFMs are computed. Figure 1 shows the evolution for the three target reactions’ efficiency rates while using FLFS-FC, studied when the number of EFMs ranges between 1000 and 20,000.

Finally, we have evaluated FLFS-FC for a large set of EFMs. In this experiment, FLFS-FC has produced a set of over 1,000,000 EFMS containing L-Lysine as a target. The evolution of the efficiency rate and the required time (in seconds) during this process can be observed in Table 2 and Figure 2.

Let us point out that, due to the random nature of the method used, it is relatively easy to parallelize the extraction of EFMs. To obtain more massive sets of EFMs, we propose to follow a task-based parallel approach, running many independent parallel tasks to extract smaller sets and, in a second step, join them while avoiding repetitions.

As a final remark, our approach can produce all the EFMs of the network (that is, any EFM could appear in the list of computed EFMs). However, the number of possible linear programs is essentially the same as the number of linear programs associated with sets of negative seeds. Therefore, a combinatorial explosion appears that needs to be handled with the use of heuristics.

## 3. Discussion

This paper shows a new way to obtain EFMs by posing LP problems. The key ingredients are choosing the right length for the objective function and suitable chosen negative constraints.

The first contribution is the study of the effect of the length of the function used on the extraction method’s efficiency.

The second contribution is the use of negative constraints to improve further the efficiency rates obtained. Due to a previous analysis of possible flux couplings, these constraints do not imply the appearance of unfeasible problems.

This way, we have developed FLFS-FC, a new approach that significantly improves efficiency while computing large sets of EFMs. Our proposal is parallelizable, so it is particularly well suited to obtaining large sets of EFMs.

As for future work, we plan to analyze the relationship among the optimal value for the length of the functions, the network’s properties, and the reaction used as a target. We also expect to slightly improve the rates obtained by choosing different sets of $I^{\prime }$ of reactions to choose the negative constraints randomly.

## 4. Material and Methods

### 4.1. Definitions and Background

A metabolic network *N* is a tuple (M,R,S,Irr) where *M* is a set of (internal) metabolites, *R* a set of reactions, *S* a stoichiometric matrix $S\in R^{M\times R}$, and Irr⊂R a subset of irreversible reactions. In this stoichiometric matrix *S*, each row represents a metabolite, and the values inside the matrix are the stoichiometric coefficients for metabolites on each reaction.

A vector of variables is defined where any of the variables gives the number of times a reaction occurs. This vector also represents the rate at which the substrate metabolites are converted to product metabolites. The vector that contains the reaction rates is called the flux rate.

A vector *v* representing flux rates of the reactions is called a mode if it fulfills the steady-state and thermodynamic constraints:(1)S·v=0
(2)v≥0
where the last condition stands for v[i]≥0 for all the components v[i] of *v*.

For any mode *v*, its support supp(v) is the set formed by those reactions *r* that appear with a non-zero rate in *v*. A mode *v* is called an elementary mode, or EFM, if another non-zero mode $v^{\prime }$ with $supp\left(v^{\prime }\right)⊊supp\left(v\right)$ does not exist (see in [6]). It is well known that the set of EFMs is finite in any network, and any mode can be written as a sum of positive multiples of EFMs (see in [5], for example).

Theorem 1 (see the demonstration in [17,22]) characterizes when a given mode is, in fact, an EFM of the network.

**Theorem** **1.**
*Given a mode v, consider the submatrix $S^{\prime }$ of S including the columns corresponding to reactions in supp(v) and rows associated with metabolites that appear in those reactions. The mode v is an EFM if and only if $rank\left(S^{\prime }\right)=\left|supp\left(v\right)\right|-1$.*


Observe that, in particular, an EFM is determined (up to positive scalar multiplication) by its support.

A reaction r∈R is said to be non-blocked if it appears in the support of some EFM. Throughout this paper, *N* stands for a given metabolic network whose *n* reactions are all irreversible and non-blocked, where *n* stands for the cardinality of *R*.

The steady-state (Equation 1) and thermodynamic (Equation 2) equations are constraints on the set *R* of reactions that define a polyhedral cone. They can be turned into an optimization problem by adding an objective function. To do so, we can use any linear function $f=\sum_{i=1}^{n}a_{i}v\left[i\right]$, where v[i] are the variables that account for the flux through each reaction *i* and ${\left(a_{i}\right)}_{1}^{n}\in R^{n}$ is a tuple of coefficients.

The associated optimization problem can be stated as
(3)$$\begin{array}{ccccc} & \mathrm{Minimize}\;\;f\\ & \mathrm{subject}\;\mathrm{to}\; & & S\cdot \vec{v}=\vec{0}\\ & \; & & v[i]\geq 0 & \;\forall r_{i}\in R \end{array}$$

Notice that in problem (Equation 3) the decision variables are the {v[i]}, and the coefficients $\left\{a_{i}\right\}$ can be thought of as parameters that allow us to pose different LP problems. We call this optimization problem a clean LP problem.

In this paper, we have always used tuples ${\left(a_{i}\right)}_{1}^{n}\in R^{n}$ with $a_{i}\geq 0\;\forall i=1,\cdots ,n$. This restriction, together with the assumption that all reactions are irreversible, ensure that the function *f* can only take non-negative values and so the posed minimization problem (Equation 3) is bounded. Observe that any solution to the above problem is a network mode, but additional conditions are needed to get nontrivial solutions.

Given two sets of reactions $T_{1},T_{2}\subset R$, the restrictions
(4)$$\sum_{i\in T_{1}}v\left[i\right]=1$$
(5)$$\sum_{i\in T_{2}}v\left[i\right]=0$$
are called, respectively, the positive and negative constraints associated with $T_{1}$ and $T_{2}$. These reactions are sometimes called positive (or negative) seeds (see in [23,24]).

By adding the positive restriction (Equation 4) to the clean LP problem, we can be sure that any solution obtained contains at least one reaction from $T_{1}$ (or the absence from the solution of all the reactions in $T_{2}$ for the negative restriction (Equation 5)). Observe that two or more negative constraints can be joined into a single constraint, but this is not true for positive ones.

### 4.2. Finding EFMs by Solving LP Problems

It is well known [25] that if we impose more than one positive constraint, not all modes with minimal support are EFMs. This is so because the convex cone is restructured with the positive constraints and so these minimal modes can be combinations of EFMs. In [25], they are called minimal constraint flux modes. The following Theorem (from the work in [26]) shows that, when restricting the number of additional positive restrictions to one, any solution obtained by solving an LP feasible problem is an EFM of the network.

**Theorem** **2.**
*Given any non-empty subset $T_{1}\subset R$ and a non-zero vector ${\left(a_{i}\right)}_{1}^{n}\in R^{n}$ with $a_{i}\geq 0$, consider the LP problem*
(6)$$\begin{array}{ccccc} & Minimize\; & & \sum_{i=1}^{n}a_{i}\cdot v\left[i\right]\\ & subject\;to\; & & S\cdot \vec{v}=\vec{0}\\ & \; & & \sum_{i\in T_{1}}^{}v\left[i\right]=1\\ & \; & & v[i]\geq 0 & \;\forall r_{i}\in R \end{array}$$
*obtained by adding to the clean LP problem the positive constraint associated with $T_{1}$. This problem is feasible, and the solution obtained is an EFM of the network.*

*Let $T_{2}\subset R$ be another subset of reactions. If we impose both the positive and negative constraints associated with $T_{1}$ and $T_{2}$, then the problem is unfeasible or the solution obtained is an EFM.*


Therefore, three ingredients are necessary to build an LP problem and compute an EFM of the network:A vector ${\left(a_{i}\right)}_{1}^{n}\in R^{n}$ with $a_{i}\geq 0$ to define the linear function *f* to be minimized.A subset $T_{1}\subset R$ to add a unique positive constraint to avoid trivial solutions.Optionally, a subset $T_{2}\subset R$ to add an optional negative constraint.

The above ingredients can be modified to pose different LP problems to obtain different EFMs of the network. Notice that this process is not straightforward because different functions and constraints can lead to the same solution.

The efficiency rate is an important figure to measure the quality of the used LP approach. This is usually defined as the quotient of the number of LPs posed and the number of different EFMs found [20]. Different strategies can be used to pose LP problems so that the efficiency rate obtained is as low as possible [18,20,27].

A first step in developing strategies is to determine what ingredient is more suitable for modification to get optimal efficiency rates. We will start with two simple strategies:
Strategy 1.—Fix the linear function *f* and vary the positive constraint. In this case:
–We will use the linear function $f=\sum_{i=1}^{n}v\left[i\right]$.At every step, we randomly choose a set $T_{1}$ of reactions and add the positive constraint associated with them.Strategy 2.—Fix the positive constraint and vary the function *f*–We will use the linear constraint $\sum_{i=1}^{n}v\left[i\right]=1$.At every step we randomly choose non-negative values ($a_{i}$) and try to minimize the associated function $f=\sum_{i=1}^{n}a_{i}v\left[i\right]$.

We have performed 50 runs of experiments which consist of the extraction of 2000 EFMs in the network model iAF1260 [21] from BIGG ([28]). The obtained (mean) efficiency rates are shown in Table 3. As observed, better efficiency rates are obtained by changing the objective function.

The obtained rates shown in Table 3 quickly increase as we try to obtain more massive sets of EFMs, so better methods are needed to lower these rates.

A related problem is to obtain EFMs having a determined set of reactions contained in its support (see in [23,29]). In the most straightforward cases (i.e., when the set of desired reactions has cardinality 1), this problem can be approached by solving the linear program problem in which the requirements are introduced as additional constraints [23]. To do so, the positive constraint must be used to ensure the appearance of this target reaction. Therefore, in the associate LP program, the positive constraint used must remain unchanged and only the objective function, and additional negative constraints can vary.

Taking into account, the results showed in Table 1 and the above remark, our proposed method is based on the random variation of the objective function and the use of additional negative constraints.

### 4.3. Enhancing the Election of Objective Functions

This subsection presents the first step to improve the extraction of large sets of EFMs. As stated above, to include the possibility of finding sets of EFMs containing a given reaction, it uses the positive constraint to ensure this target reaction’s appearance in the support of the solution.

Our strategy’s primary goal is to find objective functions that are different enough to lead to LP problems with the least possible number of repeated solutions.

The key concept is the study of the influence of the returned value in a minimization problem. This is accomplished in the following result.

**Theorem** **3.**
*Let J be a subset of R and r∈R\J. Consider the following LP problem*
(7)$$\begin{array}{ccccc} & Minimize\;f(v)=\; & & \sum_{j\in J}^{}v\left[j\right]\\ & subject\;\mathrm{to}\; & & S\cdot \vec{v}=\vec{0}\\ & \; & & v[i]\geq 0 & \;\forall i\in R\\ & \; & & v[r]=1 \end{array}$$

*If f(E)=0, then the set of possible extreme solutions of this problem is $\{E^{\prime }\;|\;E^{\prime }$ is an EFM, $r\in supp(E^{\prime })$ and $supp(E^{\prime })\cap J=\emptyset \}$*

*Given an EFM, E, of the network, it is always possible to find a subset J and a reaction r such that E is an extreme solution of the posed LP problem with f(E)>0*



**Proof.** Observe that, for any solution E′, f(E′)=0 is equivalent to saying that supp(E′)∩J=∅ and, due to the additional constraint imposed, we must have r∈supp(E′). Finally, by using Theorem 2, we obtain that all possible extreme solutions of the problem are EFMs.For any reaction r∈supp(E), define J=R\supp(E)∪{r}, and the additional constraint v[r]=1. We have
f(v)=∑j∈Jv[j]=v[r]+∑i∈R∖supp(E)v[i]Clearly f(E)=1>0 and this is the optimal value for *f* (for any other solution of the problem *v*, we must have v[r]=1 and so f(v)≥1). Furthermore, *E* is the unique minimal solution, because any other solution *v* must accomplish that supp(v)⊂supp(E) and E is an EFM. Therefore, *E* is an extreme solution of the LP problem. □

**Remark** **1.**
*This theorem is related to the well-known concept of cut set introduced by Klamt in [10]. Recall that, given a target reaction r, a set of reactions ζ is called a cut set for the reaction r if the deletion of these reactions from the network forces the reaction r to appear with zero flux in all the modes of the (modified) network.*

*Observe that, if E is the solution obtained for the problem posed in the theorem, then the condition f(E)>0 is equivalent to saying that the set J is a cut set for the reaction r.*


The first statement of the above result shows that, after fixing the reaction *r*, the functions associated with sets *J* that are not cut sets for *r* usually lead to LP problems with non-unique solutions. Moreover, those solutions are likely to appear in several different LP problems. The second statement shows that it is not restrictive to use functions from cut sets for the reaction considered.

Combining both statements, Theorem 3 suggests using sets *J* of reactions that are likely to be cut sets for the reaction *r*. On the one hand, is better to use sets *J* large enough so that they are more likely to be cut sets for the reaction *r*. On the other hand, this size should not be too large to allow the construction different sets *J* with this cardinality.

For a given network *N*, our approach starts by running several experiments consisting of finding small sets of EFMs using functions constructed from sets of reactions of different sizes. After doing this, we can select the appropriate size as the one who produces the smallest proportion of repeated solutions. In the case studied in Section 2, these experiments led us to fix the size of around 20% of the network’s number of reactions, but this proportion could vary from one network to another. Further research is left for future work.

After computing this appropriate size, our proposed strategy is accomplished as follows.

1.Starting with the target reaction *r* we calculate k=p×n, where *n* is the number of reactions and *p* the proportion of reactions used to compute the objective functions. A list to store the supports of the EFMs computed is defined.2.In each step, we randomly generate a list *J* of reactions of length equal to *k*. This list defines a linear function *f* whose coefficients correspond to indices that are in *J* and are set equal to random numbers between 0 and 1. The remaining coefficients are set to 0.3.We pose the LP problem of minimizing *f* with the stoichiometric and thermodynamic constraints and also add the constraint v[r]=1.4.Let *E* be the solution returned by the solver. We then compute its support and, if it has not been previously computed, it is added to the list of the obtained supports.

The following Algorithm 1 summarizes our approach.
**Algorithm 1:** Algorithm to extract EFMs from a metabolic network
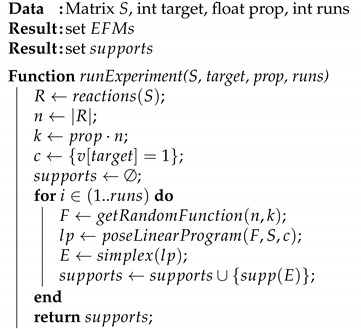


The parameters in this algorithm are the stoichiometric matrix *S*, the reaction target used to construct the positive constraint, the proportion of reactions in *R* to be included in the objective function, and the number of runs. Notice that the optimal value of prop can be obtained by running this algorithm with different values of this parameter and choosing the one more suitable for the metabolic network under study.

### 4.4. Improving the Efficiency Rates Using Negative Constraints

Negative constraints can be added to improve the efficiency rates obtained. Given a reaction s≠r, a negative constraint s=0 is added to Equation (Equation 3) in this way:(8)$$\begin{array}{ccccc} & \mathrm{Minimize}\; & & \sum_{i=1}^{n}a_{i}\cdot v\left[i\right]\\ & \mathrm{subject}\;\mathrm{to}\; & & S\cdot \vec{v}=\vec{0}\\ & \; & & v[r]=1\\ & \; & & v[s]=0\\ & \; & & v[i]\geq 0 & \;\forall r_{i}\in R \end{array}$$

We use flux coupling ([30]) to avoid unfeasible LP problems. Given two reactions *r* and *s*, it is said that *r* implies *s* if for every mode *E* of the network in which *r* appears with non-zero rate, this is also true for *s*, i.e., it also appears with a non-zero rate. This is denoted by r→
*s*.

The problem posed in (Equation 8) is feasible iff *r* does not imply *s*. Therefore, the set of flux couplings for the target reaction *r* is first computed. This can be done, for example, with the algorithm proposed in [31].

Let *I* be the set {s∈R|r↛s}, that is, the set of reactions not implied by *r*. The key idea is that, by choosing different subsets of reactions K⊂I, we should obtain different solutions by adding the associated additional negative constraints and solving their corresponding LP problems.

In order to improve the diversity of solutions computed, we propose the replacement of *I* by a subset $I^{\prime }\subset I$ containing those reactions that appear with higher frequencies in solutions of previously posed LP problems. This set of reactions $I^{\prime }$ can be obtained by using the previously calculated EFMs to get the optimal value for the length of the objective functions used.

Finally, for the chosen target reaction *r*, we estimate the best possible value for the cardinality of the subsets $K\subset I^{\prime }$ that we are going to use as additional negative constraints. This value is obtained by running several experiments of our Algorithm 2 using different values of the parameter size and choosing the one who produces better results. We want to remark that the estimation of the parameters needed for our algorithm only have to be calculated once and the time required is quite small compared with that of the process of computing large sets of EFMs.

Algorithm 2 shows our improved approach (called FLFS-FC, Fixed Length Function Sampling with Flux Coupling) in which we have another two parameters $I^{\prime }$ and size that correspond to the set of reactions that appear more frequently in the first obtained EFMs and the optimal size for sets of reactions to be considered as negative constraints.
**Algorithm 2:** FLFS-FC algorithm to extract EFMs from a metabolic network
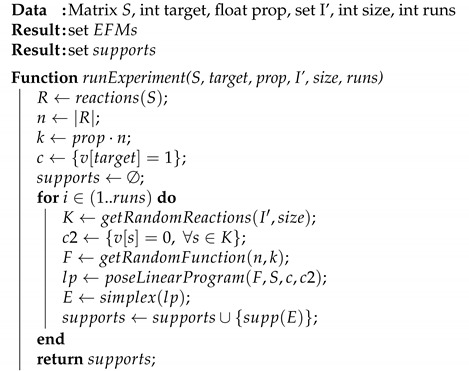


## Figures and Tables

**Figure 1 metabolites-10-00489-f001:**
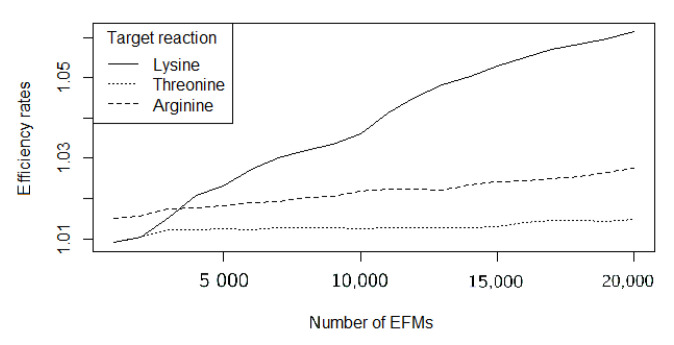
Evolution of the efficiency rates for L-Lysine, L-Threonine, and L-Arginine for FLFS-FC.

**Figure 2 metabolites-10-00489-f002:**
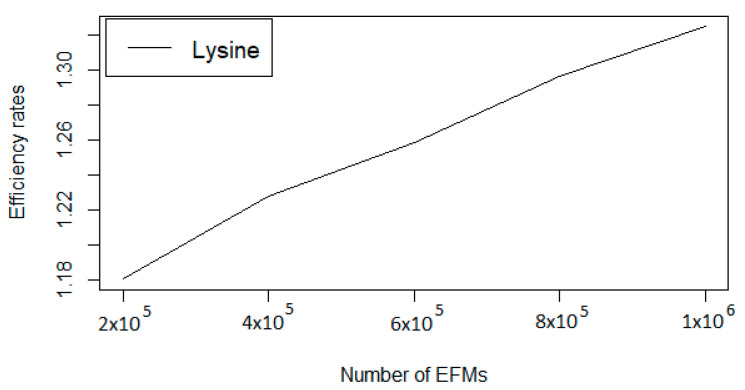
Evolution ofthe efficiency rate for L-Lysine obtained with FLFS-FC.

**Table 1 metabolites-10-00489-t001:** Comparing FLFS-FC to the other approaches for the extraction of 2000 EFMs.

Compound	EFMevolver	treeEFM	FLFS-FC	|I${}^{\prime }$|	|K|	Relative Improvement
L-Lysine	2.23	1.38	1.017	70	32	22.35
L-Threonine	1.90	1.64	1.009	70	4	66.66
L-Arginine	1.80	1.67	1.014	70	3	47.85

**Table 2 metabolites-10-00489-t002:** Evolution of the efficiency rate and computation time (in seconds) while computing one million EFMs passing through L-Lysine.

Number of EFMs	Efficiency Rate	Computation Time
20,000	1.0611	782.6315
200,000	1.1806	8044.6168
400,000	1.2277	16,972.4034
600,000	1.2583	26,785.2445
800,000	1.2964	37,428.8663
1,000,000	1.3246	48,881.2113

**Table 3 metabolites-10-00489-t003:** Median efficiency rates obtained by obtaining 2000 EFMs in the network model iAF1260.

Strategy	Efficiency Rate
Randomizing additional constraint	8.08
Randomizing objective function	1.54

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
