# Peer review of "Flux Coupling and the Objective Functions’ Length in EFMs"

_metabolites, 2020, doi:10.3390/metabo10120489_

Round 1
Reviewer 1 Report
The method described in this paper appears to be a useful improvement to available methods for rapidly generating representative samples of elementary flux modes.
I have one major comment, which does not undermine the results, but does relate to the implementation of the algorithm. In section 3.3, p.8, where negative constraints are introduced, there is reference to flux coupling, citing reference 5 for the concept and reference 18 for the algorithm. This represents the reinvention and renaming by the FBA community of the concept of enzyme subsets introduced by Pfieffer et al and Schuster (Bioinformatics, 15, 251-257, 1999) in the description and implementation of their program Metatool. As made clear in that paper, the enzyme subsets are simply extracted from the null space of the stoichiometry matrix. Furthermore, the reactions of a subset can be combined into a single overall reaction to give a reduced stoichiometry matrix that can then be used for the computation of the elementary modes; this was implemented in Metatool (and EFM software in my group), and at the end of the calculation, the elementary modes containing enzyme subsets were expanded back by insertion of their component reactions. This is not merely a point about priority of ideas; it has implications for this paper. If the present authors’ algorithm similarly began by reducing the stoichiometry matrix, then any reaction not included in the positive constraints could be safely included in the negative constraint without further checking.
From observations in my research group, determining the null space and enzyme subsets even of a genome-scale metabolic model is a rapid calculation and reduces the size of the stoichiometry matrix by about half. This would likely give a significant speed up to the repeated LP calculations in the authors’ method.
As a minor point, the ‘blocked reactions’ mentioned on p. 2, are essentially the inconsistent enzyme subsets defined by Pfieffer et al and reported by Metatool, and that paper would be a more appropriate citation than [26]. Reference 26 should be the citation for the second paragraph of p.2 rather than 25.
Reviewer 2 Report
The authors presented a method which aims at computing elementary flux modes for large networks. The method is based on Linear Programming and modified the length of the objective functions with the negative constraints to improve the efficiency rate of generation of EFMs.
I think the paper could be suitable for the journal but not in its actual form.
The authors used many variables they don't explain or later in the paper, it makes the paper difficult to read. Moreover, their algorithm used a lot of parameters which have to be justified and also tested/discussed with regard to the results.
1-
* For a better understanding, the authors should say in the introduction what the objective function represents. Because it is unclear to understand what is the length of the objective function before the method section page 4.
* The authors emphasized their method to compute few redondant pathway but why don't they use the EFMs solution as negative constraints to prevent the redondancy of the EFMs? This is the method used in logic programming with SAT solver.
page 3- what is I' ? How the set I' is chosen ?
2- THM1 correct S_v with S'
3-
* Give a formal definition of N
* The authors should explain a little the meaning of f
* When a optimization problem is written, one have to explicit the decision variables. What are the decision variable ? Does a_i are decision variables otherwise what are their values ?
* The equation (4) and (5) should have different set T otherwise it can make confusion as it results no solution.
* The problem of having more than one positive constraints is discussed in :
Minimality of Metabolic Flux Modes under Boolean Regulation Constraints (Morterol M., Dague P., Peres S., Simon L.) In Proceedings of the 12th International Workshop on Constraint-Based Methods for Bioinformatics (WCB'16), 2016. The set of minimal supports is not necessary a set of EFMs as the convex cone is restructured with the positive constraints. They can be combinations of EFMs and are called minimal constraint flux modes.
*Table 1 is unclear. What are the additional constraints ? how many test did the authors make ?
*Both algorithms have to be more detailled. Data is declared with a matrix S but the function FLFS used 2 inputs : S and number. what is number ? why prop =0.2 ? i is not initialized before the loop for, what is the value of R[i] ?
* In table 2, the method is compared with EFMevolver and treeEFM. The previous methods have to be presented in the introduction or in background.
4-
* the efficiency rate definition has already been given page 5. Not usefull page 10.
* It should be appropriate to complete the table 3 with the computing time.
* The authors didn't not mentioned the optimization solver they used.
Round 2
Reviewer 2 Report
The manuscript has been improved and the authors have responded to my comments. However, I have few remarks:
* The author argued well the objective function in the introduction. Generally the biomass function reprensent the growrh rate, the maintenance or many biological functions. I would suppress the following sentence : "The objective function has no exact biological meaning, and it is just a mathematical tool created by choosing the coefficients of the different reactions."
Author Response
We want to thank again the reviewer for their feedback. We have followed the advice and suppress the phrase pointed. We have also restyled the highlighted text in red back to black. We hope the final document meets their expectations.